# Educator perceptions of the complex needs of young people in Pupil Referral Units: An exploratory qualitative analysis

Dennis Kaip[1], Nigel Blackwood[1], Sarah Kew-Simpson[1], Alice Wickersham[2], Joel Harvey[3], Hannah Dickson[1] *

1 Department of Forensic and Neurodevelopmental Sciences, Institute of Psychiatry, Psychology & Neuroscience, King's College London, London, United Kingdom, 2 Department of Child and Adolescent Psychiatry, Institute of Psychiatry, Psychology & Neuroscience, King's College London, London, United Kingdom, 3 Department of Law and Criminology, Royal Holloway, University of London, London, United Kingdom

* Hannah.dickson@kcl.ac.uk

## Abstract

### Background

Alternative education provision such as Pupil Referral Units support young people who have been excluded from mainstream school settings and often from disadvantaged backgrounds. However, there is limited research to date exploring educators' perceptions of the complex needs of young people in PRUs, and the extent to which PRUs as currently configured can meet such needs.

### Methods

Between March 2019 and October 2020 twenty-two participants holding various educational roles from five different Pupil Referral Units across London and Southeast England were interviewed. The interviews aimed to explore the participants' experiences of working with students in PRU's and examine some of the challenges that they might encounter. Semi-structured interviews were analysed using Reflexive thematic analysis.

### Results

The three identified themes and their sub-themes highlighted the complex needs of these young people and identified significant barriers to effective service provision. The first theme 'Complexities of PRU population' highlighted the challenges that young people in PRUs face and perceived systemic short falls in addressing such complexity. The second theme 'Challenges of the PRU environment' highlights the frustrations that educators experience when it comes to providing adequate support to young people in PRU's, the absence of agency support, and the uncertainty that these educational settings can bring. The third theme 'Peer Group Influences' highlights the impact of peer groups from beyond the classroom on engagement within the classroom.

approval to do so. Data are available from the Kings' College London Ethics Committee (rec@kcl. ac.uk) for researchers who meet the criteria for access to confidential data.

**Funding:** The author(s) received no specific funding for this work.

**Competing interests:** The authors have declared that no competing interests exist.

## Conclusions

Despite the clear complex needs of young people in PRUs, staff reported feeling ill-equipped to support these individuals and lacked access to effective inter-agency support. Participants reported that pupils' mental health difficulties were exacerbated by exclusion and reintegration practices, an over-zealous focus on educational outcomes and the impact of gang influences on their school lives. Implications include more specific mental health training for staff working in PRU's, improved inter-agency working and the incorporation of trauma-informed approaches in educational practice.

## 1 Background

Alternative education provision (AP) is education arranged by local authorities for pupils who, because of exclusion or illness, would not otherwise receive a suitable education. Such provision includes Pupil Referral Units (PRUs) which support young people who have been identified as being at risk of low educational attainment and permanent exclusion in mainstream school settings [1]. Many young people attending PRUs have experienced exclusion, a common disciplinary practice used to control problematic behaviour in mainstream education settings [2, 3]. Government statistics indicate that school exclusions are on the increase, with England reporting the highest rates in the UK resulting in a crisis in demand for PRU places [4].

Young people placed in PRUs are among the most disadvantaged in society as they are at high risk of low educational attainment, unemployment and criminal offending [5]. The main goal of a PRU is to offer alternative education to students for a short duration with small class sizes and intensive support in an effort to successfully reintegrate these students back into mainstream education [6]. In reality, reports suggest that the majority of young people stay in PRUs for up to two years [7]. Common problems experienced by PRU pupils include special education needs (SEN), social, emotional and mental health issues, adverse childhood experiences including abuse and neglect, and periods of local authority care [8–12]. Accordingly, 'therapeutic' PRUs have formal educational psychology or Child and Adolescent Mental Health workers embedded in the multi-disciplinary educational team to assess and support students with more complex needs [13]. But concerns have been expressed about the quality of PRU service delivery [13], and the disproportionate use of punitive measures within some PRU settings including the application of physical restraint and use of isolation [14].

Our previous work exploring clinicians' perceptions of the mental health needs of young people in AP settings suggested that pupils are not receiving adequate mental health support or fragmented care due to shortcomings within AP schools and numerous external barriers to care [15, 16]. Given the associations between young people's complex needs like mental health, trauma, neurodevelopmental and behavioural problems and later poor outcomes like low academic attainment, unemployment and offending receiving prompt support from appropriate secondary health care or social care services is essential [17–19]. However, for example, increases in referrals into Child and Adolescent Mental Health Services (CAMHS), and evidence showing that a quarter of all such referrals are rejected [20] suggest that this might not be the case. Understanding the complex needs of this vulnerable population is a research priority with findings having implications for informing policy and practice changes.

Educators in PRUs thus face significant challenges in maintaining discipline and appropriately managing the complex needs of young people in PRUs [21]. For example, they typically

have very little specialist training in relation to young people's mental health needs and how it might impact educational engagement [22]. Yet, there is a paucity of research investigating the perceptions of educators with regards to the specific needs of young people in PRUs. This is important because of concerns over the quality of AP deriving from individual (unmet mental health needs, challenging behaviour, gang involvement) and systemic (lack of staff training, staff shortages, the use of physical restraint and isolation) factors [16]. The objective of the present study was to interview educators who work in PRUs to determine their understanding of the key needs of those young people taught in such settings, and the extent to which PRUs as currently configured can meet such needs. The study aimed to answer two research questions: (1) What unmet needs, if any, do young people in PRUs have? and (2) Is the current service provision in PRUs adequately meeting the needs of these young people?

## 2 Method

In this qualitative study, 22 participants were interviewed from five different PRUs across London and Southeast England between March 2019 and October 2020. Semi-structured interviews were conducted using open-ended questions, which are deemed suitable to examine an individual's experiences of complex and sensitive matters [23]. Reporting follows the Consolidated criteria for reporting qualitative research (COREQ) checklist [24].

### 2.1 Recruitment and participants

Participants (n = 22) were adults over 18 years who were employed in different roles within PRU's who all had direct student contact. Twelve participants were male (mean age 44.2 years) and 10 were female (mean age 41.5 years). The recruitment of staff was conducted via purposive sampling, snowball sampling and chain referral sampling through two prior identified gatekeepers embedded in PRU services. The participant cohort reflected the education professions that staff in PRU settings occupy and included roles such as teachers, teaching assistants (TAs) and referral managers. Participants had experience in working in education settings ranging from 1–26 years (mean 12 years) and had worked in PRU's for 1–15 years (mean 5 years). Participants were from five PRUs and three different organisations. One PRU was a therapeutic school for young people of secondary school age supporting their mental health needs. The other four PRUs worked with children and young people excluded from primary and secondary mainstream schools and all offered short- and long-term placements. See Table 1 for details of participant gender and job role. Ethical approval for the study was obtained from the university research ethics panel (RESCM-19/20-9231).

### 2.2 Measures and materials

The interview schedule was developed by authors DK and HD following agreement on the study research questions (see S1 File for a copy of the interview schedule). During the interview with author DK participants were asked to share their experiences of working with students in a PRU and to identify the challenges and barriers encountered by staff and students. Interviews were recorded using the "VoiceRecorder" audio app on an iPad.

### 2.3 Procedure

Individuals identified were contacted via email and invited to participate in the study. A suitable time and location for the interview was then arranged. All interviews were conducted in person in locations such as classrooms, school offices or participant residences. Participants were informed that they could stop the interview at any stage without providing a reason.

**Table 1. Job roles of participants.**

| ID | Sex | Occupation |
|----|-----|------------|
| 1 | Female | Teacher |
| 2 | Male | Head Teacher |
| 3 | Male | Outreach Support Worker |
| 4 | Female | Referral Manager |
| 5 | Female | Teacher |
| 6 | Male | Deputy Head Teacher |
| 7 | Male | Assistant Head Teacher |
| 8 | Male | Teaching Assistant |
| 9 | Male | Head Teacher |
| 10 | Female | Learning Support Assistant |
| 11 | Female | Learning Mentor & Parent Instructor |
| 12 | Female | Designated Safeguarding Lead |
| 13 | Female | Admin & Student Support |
| 14 | Male | Learning Support |
| 15 | Male | Assistant Head Teacher |
| 16 | Male | Deputy Head Teacher |
| 17 | Male | Service Instructor |
| 18 | Male | Learning Mentor |
| 19 | Female | Learning Support Coach |
| 20 | Female | Head Teacher |
| 21 | Male | Deputy Head Teacher |
| 22 | Female | Exam Officer & Teacher |

Additionally, participants provided written consent to take part in the study and were informed about their right to withdraw their consent to participate at any point up to the end of December 2020 without providing a reason. Demographic information was collected, followed by the semi-structured interviews ranging from 19 minutes to 56 minutes with a mean duration of 33 minutes. Once the interview was completed participants were thanked for their time and reimbursed £20. Interviews were audio recorded and transcribed verbatim for further analysis.

## 2.4 Data analysis

Due to the exploratory nature of the study, reflective thematic analysis was deemed to be the most appropriate method to examine the research questions [25] and to establish a clear understanding of participants' thoughts and experiences [26]. An essentialist framework was employed and an inductive approach was used to identify themes on a semantic level. Each transcription underwent the same data analysis process outlined by the six-phase approach in thematic analysis [25]. During the process, a connection between identified codes and generated themes across all data sets was established. Overarching themes were collated on an initial thematic map before further distilling those into final themes and subthemes. These were then refined and updated on a thematic map. To ensure that the quality and credibility of the results were upheld, all transcripts were analysed by the first author (DK) and independently checked by one of the co-authors (JH). First author (DK) was an MSc student at the time the study took place but was an experienced qualitative researcher and had previously worked in youth services. Author DK undertook the data analysis under the supervision of authors JK and HD.

### 2.5 Reflexivity

While this study used an inductive approach, the researcher engaged in internal dialogue and critical self-evaluation. These reflections were recorded in a reflection log. The researcher also engaged in regular, reflective discussions with the research team.

## 3 Results

Analysis of the 22 semi-structured interviews elicited three themes and nine sub-themes. The first theme 'Complexities of PRU population' highlighted the challenges that young people in PRUs face and perceived systemic short falls in addressing such complexity. The second theme 'Challenges of the PRU environment' highlights educator frustrations and uncertainties in such settings. The third theme 'Peer Group Influences' highlights the impact of peer groups from beyond the classroom on engagement within the classroom. Fig 1 shows the three themes and sub-themes.

### 3.1 Complexity of PRU population

Across all PRU settings participants recalled experiences with young people that went beyond challenging behaviours and special needs presentations that are commonly present within a typical PRU setting. Despite most participants describing complex needs, the support that was available for young people and staff alike in understanding and remediating such needs was limited.

**3.1.1 Mental health challenges.** Mental Health Challenges were identified and experienced by most participants across all PRU settings. It was suggested that many young people in the PRU had behavioural difficulties in the context of co-morbid mental health conditions. While conditions such as low self-esteem, anxiety and depression were recognised by most participants, post-traumatic stress disorder (PTSD), attention deficit hyperactivity disorder (ADHD) and severe mental illness were also apparent:

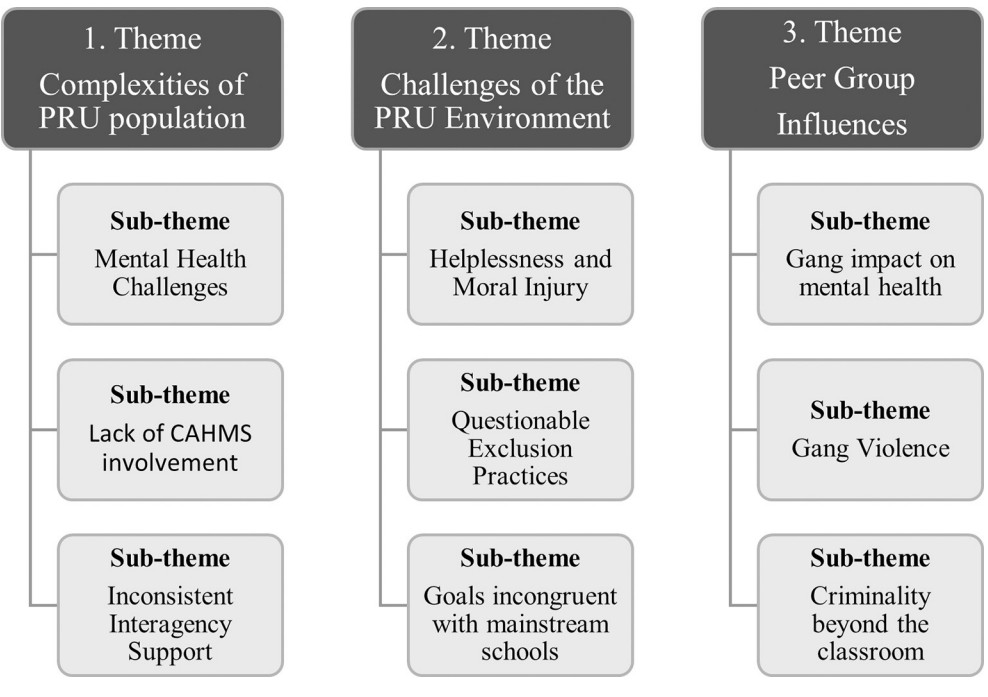

**Fig 1. Themes and sub-themes.**

"*Autism spectrum, PTSD, ADHD, I've seen foetal alcohol syndrome in PRUs. . .*" *(P-17)*

"*. . . everything from paranoid schizophrenia, suicidal ideation, eating disorders neurodevelopmental issues, to autism, ADHD, the full gamut.*" *(P-21)*

The intensity and impact of major mental health difficulties were illustrated by participant 2 who described an event with a student in year 11 (15–16 years). This student had several mental health diagnoses and had previously been arrested for setting fire to a major train station in London:

"*. . .the worst incident was when she came into the building and she just started. She had a bottle of WKD which she clearly been drinking [. . .] She smashed every window down the corridor. . .she smashed wooden chairs into firewood. We had evacuated all the other pupils and members of staff. . . she had smashed the bottle at that point and she was cutting, she was gouging it into her hand.*" *(P-2)*

Several participants shared similar experiences of serious incidents including suicidality which were not isolated occurrences:

"*One of our students who left us is 18, he was 18. He committed suicide about a month ago [. . .] he was such as lovely boy [. . .] and was doing so well for such a long time*". *(P-4)*

"*We have children who are on suicide watch and they have to. (. . .) the young lady who I was talking to you (about), (. . .) a member of staff must be with her at all times*". *(P-21)*

Despite the reported severity and complexity of mental health presentations, most participants reported that pupils did not receive adequate mental health support or questioned the current support available.

**3.1.2 Lack of CAMHS involvement.** Most participants agreed that CAMHS input for students is essential. However, in all PRU settings it was reported that the input from CAMHS was insufficient, and it was unclear how to obtain the required support, as participant 11 explained:

"*We can do the referral, however, I will suggest that she takes him to the doctor and it's quicker. Because I can do a referral, but it takes quite a long time [. . .] I think the majority of our young people need some sort of CAMHS input. [. . .] You can do a referral but then they would say (no).*" *(P-11)*

"*For CAMHS workers, it's just paperwork and I think sometimes we're just getting lost in paperwork and we're just not seeing human beings, spending enough time.*" *(P-15)*

This was also confirmed by participants from PRUs that had more regular access to CAMHS workers. The relationship between PRU and CAHMS services was described as complex and the input was not always as beneficial as anticipated.

"*. . .a lot of our children [. . .] they don't just come from this borough [. . .]. You might be dealing with 6 different CAMHS across London which is really difficult. . .. when CAMHS suggest things to do, most of the time we're doing it anyway.*"*(P-5)*

"*I constantly contact CAMHS. The long and the short of it is, in this borough CAMHS, unless a child is about to throw themselves off a building, the waiting list of CAMHS is over a year.*" *(P-12)*

The lack of CAMHS support was also noted by a participant in the therapeutic PRU. All participants from a therapeutic PRU shared concerns about recent developments where specialist mental health staff had been taken out of the PRU. This had been detrimental to both the young persons and the overall service provision.

"...we used to have an onsite CAMHS worker that used to be here full-time, but they've actually taken him away now [...] which is, as you can imagine, it's pretty tricky, especially with the kids that are sort of more higher tier mental health issues. [...] if something goes wrong upstairs or someone self-harms or doesn't there's no one really to sort of cope with the more severe mental health students." (P-22)

This subtheme illustrates how the service provided by CAMHS was thought to be insufficient by staff in PRUs, indicating that recent NHS budgetary constraints have negatively impacted on inter-disciplinary work and support.

**3.1.3 Inconsistent interagency support.** The sub-theme "Inconsistent Interagency Support' highlights how the gap in mental health support caused by inadequate CAMHS input was exacerbated by additional difficulties in inter-agency working. Social services were named as one of the most inconsistent services that did not necessarily operate in the best interests of the young person, particularly given that PRUs have a high number of young people with histories of local authority care and who are vulnerable to child sexual exploitation, social service input was expected:

"I haven't had particularly good relationships or situations with social workers. [...] some of them seem to be desensitised to what's happening to some of these children and when we phoned up some social workers with safeguarding issues they were kind of very, very blasé about it..." (P-6)

This was confirmed by the experiences of participant 22 who shared having difficulties getting in contact with social workers on urgent matters:

"I find social workers a little bit slow at times. They don't get back to you very quickly [...] if there's a desperate sort of MASH (Multi-Agency-Safeguarding-Hub) referral or anything and you need someone urgently, it is quite hard to get hold of them sometimes". (P-22)

However,
Interestingly, the majority of participants noticed a positive change of approach from the police who were considered to be helpful by young people and staff:

"The police at this point in time are very focussed. In terms of [...] not criminalising young people, [...] I think that the police communication and partnership working is effective at the moment." (P-9)

Some participants noted that members of the police make additional efforts to try and improve the relationship with the young people in the PRU:

"The Police Officer especially has a really good understanding of our students and he comes without his uniform on. [...] because he is not in uniform the children are not confrontational with him [...] they chat to him and joke with him". (P-10)

Despite the evident need for external mental health support, most participants stated that support from child and family third-sector organisations had been infrequent and was not available in all settings. Additionally, where external mental health support was offered by professionals, their input was deemed unclear or unhelpful.

*"The psychotherapist told a member of my staff that the child was just manipulative but didn't feel able to turn around and, you know, tell the parents exactly what he thought. He felt that he almost needed to come up with something to justify the child's behaviour." (P-6)*

### 3.2 Challenges of the PRU environment

Most participants felt a sense of inadequacy when dealing with the complex difficulties of young people. All participants agreed that students were not always appropriately placed in a PRU and that the reason for their transfer to a PRU was a cause for concern. Several factors were identified as potentially contributing to the inappropriateness of their placement in PRUs.

**3.2.1 Helplessness and moral injury.** Feeling unable to deal with complex presentations and extreme situations such as suicide and serious assaults were shared by most participants. Although some participants stated that they had received training on restraint, they had not received support to manage complex mental health presentations of pupils or their emotional well-being effectively.

*"I haven't had enough training on mental health yet I'm dealing with it day in day out and I don't think that's good enough for these kids". (P-5)*

Participant 18 highlighted that dangerous situations occurred frequently:

*"I've watched someone die before from a stab wound just outside and stuff like that. And even myself just being approached and having knives pulled out on me". (P-18)*

Being aware that the challenges of working in a PRU can have an impact on the emotional well-being of staff, participant 1 expressed the view that:

*"I do think they do need to make sure there is more care for the teachers actually because I think the teachers in a PRU give a lot emotionally [. . .] you're taking on secondary trauma or something [. . .] somehow it does maybe stay with you." (P-1)*

The lack of training and supervision in addition to traumatising experiences experienced by staff regularly resulted in them feeling overwhelmed and unable to cope with young people effectively.

*"I think sometimes people think the children are in here because people been too soft and they never had any consequences for what they've done and if you're a lot, a lot harder with them then they probably won't do what they do. So it's breaking that myth amongst all staff. . . ." (P-6)*

**3.2.2 Questionable exclusion practices.** All participants stated that in some instances the decision to exclude young people from mainstream education appeared questionable. Reasons included challenging behaviours, the socioeconomic status of the family and the desire to

maintain mainstream school rankings. Participants questioned the approach adopted by mainstream education toward the young people who presented with challenging behaviours:

"*A lot of them have been threatened at their mainstream school, 'if you don't sort yourself out, I will send you to a PRU'. Threatened by other professionals, threatened by their teachers in mainstream. [. . .] 'we'll send you to a PRU as a punishment. . .'*" (P-2)

"*I do think it really, really hurts a child when they are told, no you don't belong in our society and this is what mainstream society does. When they exclude, permanently exclude a child, it is really almost educational suicide [. . .]. They feel like something has died in them and to get them built up again, isn't an easy thing.*" (P-9)

Participants 6 and 16 provided further examples of young people who had been excluded for questionable reasons.

"*His mother committed suicide. Two weeks after his mother committed suicide, he was permanently excluded saying that his behaviour had become extreme.*" (P-6)

"*. . .we've got a child at the moment [. . .] definitely got traits of autism and he was excluded for rude behaviour [. . ..] He shouldn't have been excluded* (P-16)

Participant 6 added that students from families that were from lower socio-economic backgrounds were particularly at risk of being transferred to a PRU:

"*. . . their families don't have any influence in society. They're kind of the easiest ones to, dare I say, to shit on and you know. [. . .] I used to work in Kensington and Chelsea, any incidents with some of their residents because of the influence of some of them, people would jump to make sure that things were sorted.*" (P-6)

Participant 6 also commented:

"*. . .some schools are willing to pay local authorities to keep them in pupil referral units [. . .] in doing that they're ensuring that these problem children [. . .] aren't gonna be using up resources of the staffing [. . .] and will make it a bit easier for the rest of the children to access curriculum and learning and therefore getting better SATs.*" (P-6)

Participants also expressed the view that some exclusions were driven by the need for mainstream schools to achieve higher rankings, or an unwillingness to jeopardise existing ranking.

"*It's easy to get rid of children. At one point it was always around, having worked in mainstream, it was always around examination time because it was easier to get rid than to put a child through an exam and then fail it*". (P-12)

"*I think that is a national picture. I think (mainstream) head teachers are under a lot of stress in terms of their results, sort of a factory that they have to get themselves through to keep their school afloat, keep the school in a good place in terms of OFSTED. [. . .] inevitably what happens, the kids that are the most needy are the ones that will be sacrificed.*" (P-9)

Overall, participants argued that exclusion practices in mainstream schools were often detrimental to many vulnerable young people and ultimately favoured the systemic needs of mainstream schools.

### 3.2.3 Goals incongruent with mainstream schools.

Most participants suggested that mainstream school settings focussed primarily on their rankings and that there was little interest in the reintegration of young people back into mainstream education. Most participants thought it was important to think beyond educational attainment to support the future well-being of young people in PRUs. Many participants considered that mainstream schools employed overly punitive approaches to dealing with young people with challenging behaviours, which were often due to underlying mental health difficulties and traumatic circumstances.

"...*why would a mainstream school throw a child out in year 11? [...] Why not let us work, going to them and let them work with us? They just seem like, they don't want to back down. You know, these teachers don't want to back down, it's proving a point. [...] You're going to ruin his whole career and he has only a few weeks left". (P-11)*

Participant 9 stressed that while collaboration between mainstream and alternative education settings was vital for the young person, in reality, it was often absent:

"*There needs to be a bridge between mainstream and alternative provision built. We can build a bridge halfway across the river, but the other side of the river, the mainstream needs to build theirs to meet us in the middle. [...] The culture unfortunately in some mainstream schools [...] is very exclusive not inclusive" (P-9)*

Participant 15 added that young people and the PRUs are set up to fail:

"*I fear that we're judged very much on academic progress, but I don't think anyone really takes into full consideration the hurdles that we have to overcome to get these children ready [...] I've found in my career that the complexity of the problems that we're now facing has grown exponentially. (P-15)*

This subtheme demonstrated that mainstream and PRU educational settings have a different understanding of the goals of a young person's educational journey. Most participants considered that young people were placed in PRU settings without an option to return to mainstream education.

## 3.3 Peer group influences

Participants across all PRU settings shared that gangs had been a consistent part of young people's lives. It was clear that gang criminality placed young people at risk of exploitation and harm while also making the PRU environment a target for those gang operations. An important factor highlighted by participants was that gangs had a significant impact on young people's mental health. Educators noted that gang affiliation could serve as a barrier to entry to therapeutically focused PRUs.

### 3.3.1 Gang impact on mental health.

Many participants highlighted that the influence of gangs negatively impacts young people's mental health. Participants indicated that new students often experienced stress and anxiety when trying to navigate the PRU environment. It was explained that young people, even those without affiliation to a gang, might be targeted by gangs. This is of particular concern when a young person is placed in a PRU outside their local area:

*"They can be at risk because [. . .] they can easily slip up and share their location by just trying to do a normal (social media) post, they let their guard down or (being) naive because some of these are in actual gangs (or) representing gangs (from) that area". (P-17)*

This hyper-vigilance leads to high levels of anxiety and stress:

*". . . when students come here that are not in the gangs [. . .] they're coming with a high level of anxiety because they don't know what to expect and they don't know what's gonna happen". (P-10)*

Participant 12 discussed the case of a 13-year-old student who experienced tremendous stress and feared for his life due to high gang activity in his PRU:

*". . . (he had) a life-size doll and he cut all the hair off it. He had slit the doll's head off with the scissors and he'd poke the eyes to the back of the head. He said that he had done that because that's what's going to happen to him in his head. This is what the gangs were going to do when they get a hold of him". (P-12)*

Participants highlighted that gangs impacted greatly on the young people's behaviour and mental health by being omnipresent in almost all aspects of a young person's stay in a PRU, with young people having to consider multiple factors to attempt to ensure their safety.

**3.3.2 Gang violence.** Almost all participants shared experiences of working with young people who had been associated with gangs by either working for them, being victimised by them or both. Importantly, while the seriousness of gang activity was highlighted it was apparent that gang culture is intrinsically linked to the PRU environment.

*". . .lots of pupils carried knives. Lots of pupils, lots of London PRU pupils are gang-involved [. . .]. I would say at some level 50% of the older boys, the year 10 and 11 boys were gang-involved on some level." (P-2)*

A serious event was shared by participant 5 who witnessed a gang-related attack on a young person just outside the PRU grounds:

*"I've witnessed a knife attack on a boy [. . .] just outside school [. . .] they dragged him into a doorway [. . .] Three boys on one. Laying into him and kind of it was all getting heated now, trying to take stuff from him and then I saw they had a knife as well and it was kind of like that moment of what on earth do you do." (P-5)*

Further insight was provided by participant 8 who witnessed a young person attempting a serious assault on another student which was assumed to be related to a young person glorifying gang culture:

*"We had a kid who had brought a screwdriver to stab another student and this kid is 9 years old. [. . .] He got a screwdriver and he planned that he was gonna do it at break time." (P-8)*

**3.3.3 Criminality beyond the classroom.** Several participants reported that some young people who have been inconspicuous during their placement in the PRU were heavily involved in drug operations, referred to as 'County Lines', or other gang activity. Participant 3 9 shared

their experiences of working with young people who seemed unlikely candidates for involvement in organised gang crime:

"...I've heard things, seen things from the police officers, the youth offending team [...] but they would always call me Sir and they would be as quiet as anything. They wouldn't cause any disruptions; they wouldn't get into any fights. But [...] the two [...] have been seriously involved, came in and they were almost like model students." (P-3)

This is aligned with comments from participant 7 who spoke about a student whom he referred to as an 'authentic one', which is a term used for young people involved in high-level criminality:

"Used to be quiet as a lamb. [...] One day he came in [...] with a cast on his arm. It turns out that he was part of some sort of gang in his area and it was like a turf war, (that's) how he got his arm shattered. And one afternoon [...] (he) saw one the kids and just started [...] a real violent assault. [...] Went straight into custody for a significant period of time." (P-7)

'County line operations' were discussed:

"...it's a professional organisation [...]. with county lines, these kids will be told, 'don't show up in school and don't get caught in school with anything, don't bring anything to the school. You're working for me right, if they call your bluff in school [...] you're in trouble.' Obviously, we have intel from the police, but we have a significant amount of kids being involved." (P-9)

"The more challenging ones are the kids who are on the streets are involved in the gangs, county lines and dealing with the drugs are involved in violence. So they're getting groomed. So that is the nature of the beast that we work with". (P-15)

## 4 Discussion

This study sought to examine educators' perceptions of young people's mental health needs in PRUs and the extent to which PRUs are able to meet these needs. A thematic analysis of 22 semi-structured interviews conducted in PRUs across Greater London and a county in southern England produced three overarching themes, namely 'Complexities of PRU population', 'Challenges of the PRU environment' and 'Peer group influences'. Overall, subthemes indicated that many young people in PRUs presented with significant mental health and neurodevelopmental issues, such as psychosis and suicidal behaviour, which PRU staff are not adequately trained to deal with. Inconsistent agency support and unclear referral processes meant that young people were often not adequately supported. Additionally, PRU staff reported that questionable exclusion and reintegration practices, an inappropriate focus on educational outcomes and the presence of gangs exacerbated pupil's mental health difficulties.

The findings from this study are consistent with existing literature suggesting that young people in PRUs often come from disrupted family backgrounds, have been exposed to domestic and/or gang related violence, and have significant untreated mental health difficulties [16, 27]. Our findings also show that for staff working in PRUs, understanding and managing their pupils' mental health needs appears to be one of the greatest challenges they face. This highlights the need for specialised training and support for educators [28, 29]. Indeed, supporting educators through training to address student mental health needs has the potential to not only improve outcomes for young people [30], but also reduce teacher burnout [31, 32].

However, in the present study, participants reported that requests for additional training and support were typically ignored, except for 'restraint training' to deal with aggressive incidents in PRUs, which can lead to physical and mental harm to the young people [33].

Participants reported that CAMHS are often unwilling to accept referrals for youth in PRUs with conduct problems. Among youth in PRUs, these behavioural problems have a complex relationship with undiagnosed mental health difficulties [16]. Moreover, the high levels of undiagnosed and untreated mental health problems reported by participants indicates that young people in PRUs do not receive support that is consistent with recent child and adolescent mental health policy, such as Future in Mind and the Five Year Forward View for Mental Health. These policies acknowledge that early intervention and swift access to mental health services is of paramount importance [20]. Mental health interventions that treat individuals when early symptoms emerge should theoretically minimise further development of symptoms and associated biological damage, improve long-term treatment response, and even outcomes [34]. However, participants in the present study reported significant difficulties when trying to navigate the complex referral processes to CAMHS. This may be because CAMHS is insufficiently resourced to support cohorts with complex presentations and backgrounds, as our study suggests, with the lack of a robust evidence base for the treatment of conduct problems in later adolescence [35]. This finding is supported by Frith's (2017) review of CAMHS transformation plans which suggests that eligibility thresholds based exclusively on diagnosis may prevent individuals with sub-threshold difficulties, behavioural problems and complex trauma histories from receiving early assessment and intervention also highlighting that those from disadvantaged backgrounds are less likely to access services [20, 36].

Participants reported that many young people in PRUs were exposed to adverse childhood experiences, family-related violence, or gang-related violence before and during their PRU placement. Evidence suggests that those who have been exposed to or witnessed adverse childhood experiences are more likely to engage in delinquency and violence during adolescence and adulthood [37, 38]. A systematic review of research from 1990 to 2015 found that traumatic event exposure and traumatic stress symptoms in students was associated with poorer academic and social-emotional-behavioural outcomes [39]. These studies highlight the need for more therapeutic, trauma-informed approaches to specialist education, such as the SECURE STAIRS framework [40], or trauma sensitive schools [41]. The need for a trauma-informed approach in PRU settings was also confirmed in our prior research examining clinicians' perceptions of the mental health needs of young people in AP educational settings [16].

One of the key elements in providing appropriate provision for young people with additional social, emotional, and educational needs is effective multiagency working amongst the professionals involved in the young people's lives [22, 42]. With the exception of the police, this study found that multi-agency working was identified to be inconsistent. This is worrying given that multi-agency support is one of the key interventions for addressing disciplinary exclusions resulting in a PRU placement [43]. In 2021 the UK government, as part of its 'Beating Crime Plan' announced plans to pilot an AP specialist taskforce, which aims to provide intensive multi-agency support to vulnerable children and young people in AP toreduce truancy, not in education, employment, and training (NEET) rates, the risk of involvement in serious violence and to improve mental health and wellbeing.

Participants shared experiences of observing young people being inappropriately excluded from mainstream schools. The long-term negative consequences of exclusion from school are well documented [44, 45]. Since the introduction of league tables by the Education Reform Act [46], young people who are academically less successful or have behavioural difficulties are considered less desirable and unsuitable for mainstream education, leading to higher rates of school exclusion and academic and social marginalisation [47, 48]. These experiences are likely

to exacerbate a young person's mental health difficulties as well as negatively impact learning and attainment [49, 50]. Our study and others also found that lower socioeconomic background of an individual and their family was an important factor when it came to mainstream schools making decisions on whether to exclude a student or not [51]. Indeed, an analysis by the Education Policy Institute in 2017 reported that young people from low-income backgrounds are at four times the risk of exclusion than other young people [52]. One of the goals of a PRU is that it is supposed to reduce the negative consequences of exclusion and enable reintegration back into mainstream education [53]. However, several participants in the current study shared that mainstream educators seem to expect PRUs to 'fix young people' before they would consider reintegration. Consequently, many young people remain in PRUs longer than necessary as reported at the House of Commons Education Select Committee in 2018 [5]. This may lead to differences in funding provided by local authorities to PRUs resulting in inconsistent quality of education and support [16].

The Office of National Statistics estimated that, in 2017, over 46,000 children in England alone were in gangs [54]. The recruitment into gangs is often well-organised and targeted toward vulnerable young people who have either been excluded from a mainstream educational setting or attended PRUs [55]. Participants reported that a substantial number of young people in their PRUs were associated with gangs as either victims or members, leading to increased violence between young people and against staff. The failure to correctly identify those at risk of criminal exploitation can lead to deleterious consequences for the young people and the wider PRU setting. Those consequences include physical and sexual abuse, psychological traumatisation, threat to personal safety and those of family and friends and criminalisation [56, 57]. Moreover, results of this study highlight that young people's mental health difficulties were exacerbated by gang affiliations. In light of these findings, closer working relationships between PRUs, Youth Offending Teams and mental health services are warranted to establish methods of identifying and protecting youth at risk of criminal exploitation.

We highlight two particular strengths of this study, First, a qualitative approach enables the collection of in-depth data and provides a comprehensive examination of staff perceptions of young people's needs. A further strength was the use of semi-structured interviews. Semi-structured interviews enable reciprocity between the researcher and the participant. This flexibility enables the researcher to create follow-up questions that better match the participant's responses, while also maintaining the direction and focus of the interview [58]. A key limitation of this study is that the interviews were conducted in a number of deprived areas across London and Southeast England. However, findings from this study should be broadly applicable to other PRU setting in comparable circumstances in other areas of the UK. This study forms part of a larger body of research using administrative data to examine the educational journeys of individuals interacting with the criminal justice system (CJS). This overarching goal may have shaped the conversations between the interviewer and staff at the PRUs particularly as young people in PRUs are more likely to be involved in the CJS compared to those in mainstream education [5]. However, individuals analysing qualitative data are integral to the final product, and 'bias' as applicable to quantitative research is not relevant to qualitative research where the researcher cannot step outside their own values or opinions [59].

## 5 Conclusion

The findings indicated that, despite the clear mental health needs of young people in PRUs, staff were ill-equipped to support young people and lacked access to effective inter-agency support. The nature of mainstream exclusion practices was called into question. The influence of gang affiliations on the child's PRU experience was highlighted. The need for more specific

mental health training for educational staff and the incorporation of trauma-informed approaches in educational practise are potential avenues for development. This study further uncovered the urgency to revisit educational exclusion policies and establish policies focussing on enabling more cohesive relationships amongst key agencies in the excluded population.

## Supporting information

**S1 File. Interview schedule.**
(DOCX)

## Acknowledgments

The authors would like to thank all the participants who took the time to take part in the study. HD and AW are affiliated with the National Institute for Health Research (NIHR) Maudsley Biomedical Research Centre at South London and Maudsley NHS Foundation Trust and King's College London. The views expressed are those of the author(s) and not necessarily those of the NIHR or the Department of Health and Social Care.

## Author Contributions

**Conceptualization:** Nigel Blackwood, Joel Harvey, Hannah Dickson.

**Data curation:** Dennis Kaip.

**Formal analysis:** Dennis Kaip, Sarah Kew-Simpson.

**Investigation:** Joel Harvey, Hannah Dickson.

**Methodology:** Alice Wickersham, Hannah Dickson.

**Project administration:** Dennis Kaip.

**Supervision:** Hannah Dickson.

**Writing – original draft:** Dennis Kaip.

**Writing – review & editing:** Nigel Blackwood, Sarah Kew-Simpson, Alice Wickersham, Joel Harvey, Hannah Dickson.

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
