## [Decision Letter · Decision Letter 0]

30 Jul 2024

PONE-D-24-17686Educator perceptions of the complex needs of young people in Pupil Referral Units: An Exploratory Qualitative AnalysisPLOS ONE

Dear Dr. Dickson,

Thank you for submitting your manuscript to PLOS ONE. After careful consideration, we feel that it has merit but does not fully meet PLOS ONE’s publication criteria as it currently stands. Therefore, we invite you to submit a revised version of the manuscript that addresses the points raised during the review process.

We look forward to receiving your revised manuscript.

Kind regards,

Cathryn Knight

Academic Editor

PLOS ONE

Journal Requirements:

**Additional Editor Comments:**

Thank you for submitting this article which needs some minor revisions before it can be accepted. In particular I would like to draw your attention to point 1, 5 and 6 on the PLOS guidelines for qualitative research:

1) defined objectives or research questions; 2) description of the sampling strategy, including rationale for the recruitment method, participant inclusion/exclusion criteria and the number of participants recruited; 3) detailed reporting of the data collection procedures; 4) data analysis procedures described in sufficient detail to enable replication; 5) a discussion of potential sources of bias; and 6) a discussion of limitations.

Please include reference to these points for your next submission.

Reviewers' comments:

Reviewer's Responses to Questions

**Comments to the Author**

1. Is the manuscript technically sound, and do the data support the conclusions?

Reviewer #1: Yes

Reviewer #2: Yes

2. Has the statistical analysis been performed appropriately and rigorously? 

Reviewer #1: N/A

Reviewer #2: N/A

3. Have the authors made all data underlying the findings in their manuscript fully available?

Reviewer #1: Yes

Reviewer #2: Yes

4. Is the manuscript presented in an intelligible fashion and written in standard English?

Reviewer #1: Yes

Reviewer #2: Yes

5. Review Comments to the Author

Reviewer #1: Thanks for this well-written article which addresses an under-researched area. The manuscript is well-written and engages with recent relevant literature. I have only some minor comments and these are all suggestions/ things to consider.

Line 36 of abstract missing apostrophe at particiapnts’

Line 118 in Methods- you have gender when I think you mean sex (as your labels are male or female)

Clear account of methodology although I’d welcome a bit more information on how the interview schedule was developed, and perhaps the schedule made available so future studies could ask the same questions.

Range of length of interview from 19 minutes to 47 seems very wide – any patterns in e.g. staff roles of the shorter interviews? Or any other reflection on this wide difference?

A bit more detail on the nature of the PRUs these staff work in would be helpful – what ages do they cater for? How many pupil places? Are they stand alone units? Some seem to be explicitly ‘therapeutic’ – what was the approach in others?

The analysis process is briefly but clearly reported – and is consistent with expectations for Reflexive thematic analysis

Mentions the researcher engaging in conversations with the research team – some clarity on who did what would be useful (e.g. did one author conduct the interviews, transcribe them and analyse them?)

The data presented help illuminate the themes which are subsequently linked to existing literature. The findings add to the overall understanding of the challenges facing this sector.

L556 there’s a comment about may not be generalisable to other PRUs in different areas….that’s true – but the criteria considered given this type of study ought to be transferability (which will also be limited for the reasons given, but the use of ‘generalisability’ suggests a different kind of research (small-scale, in -depth, qualitative) – though I note the ‘essentialist’ approach to RFA which does point to a realist ontology so I know it isn’t a straightforward distinction, but I still found the ‘generalisability’ a bit jarring.

Reviewer #2: Scientific article fits all the criteria.

However, the are some unclear cases, for example, 1) the table is separated from its title; 2) The sentence is missed in 251 line, isn't it? 3) Whose participant (3 or 9) thoughts are there (426 line) share in the text?

6. PLOS authors have the option to publish the peer review history of their article (what does this mean?). If published, this will include your full peer review and any attached files.

Reviewer #1: No

Reviewer #2: No

---

## [Author Response · Author response to Decision Letter 0]

22 Aug 2024

Academic Editor:

I would like to draw your attention to point 1, 5 and 6 on the PLOS guidelines for qualitative research:

1) defined objectives or research questions; 2) description of the sampling strategy, including rationale for the recruitment method, participant inclusion/exclusion criteria and the number of participants recruited; 3) detailed reporting of the data collection procedures; 4) data analysis procedures described in sufficient detail to enable replication; 5) a discussion of potential sources of bias; and 6) a discussion of limitations.

RESPONSE: We thank the academic editor for reminding us of the PLoS guidelines for reporting qualitative research. In the revised manuscript we have included our research questions. See below for research questions included at the end of the introduction:

“The study aimed to answer two research questions: (1) What unmet needs, if any, do young people in Pupil Referral Units (PRUs) have? and (2) Is the current service provision in PRUs adequately meeting the needs of these young people?” (lines 103-105)

In the discussion we now include a brief discussion on sources of bias and elaborated on our discussion of the study limitations. See below for additions to the discussion section. 

“This study forms part of a larger body of research using administrative data to examine the educational journeys of individuals interacting with the criminal justice system (CJS). This overarching goal may have shaped the conversations between the interviewer and staff at the PRUs particularly as young people in PRUs are more likely to be involved in the CJS compared to those in mainstream education (5). However, individuals analysing qualitative data are integral to the final product, and ‘bias’ as applicable to quantitative research is not relevant to qualitative research where the researcher cannot step outside their own values or opinions (58).” (lines 572-579)

We can also confirm that the revised manuscript complies with PLoS One style requirements and that our reference section is complete. As recommended, we used the COREQ to ensure the comprehensive reporting of qualitative studies, which we acknowledge in the methods section (line 111-112). Additions to the methods based on the COREQ are below:

“The interview schedule was developed by authors DK and HD following agreement on the study research questions (see supplementary materials S1 for a copy of the interview schedule). During the interview with author DK participants were asked to share their experiences of working with students in a PRU and to identify the challenges and barriers encountered by staff and students. Interviews were recorded using the “VoiceRecorder” audio app on an iPad.” (lines 133-138)

“To ensure that the quality and credibility of the results were upheld, all transcripts were analysed by the first author (DK) and independently checked by one of the co-authors (JH). First author (DK) was an MSc student at the time the study took place but was an experienced qualitative researcher and had previously worked in youth services. Author DK undertook the data analysis under the supervision of authors JK and HD.” (lines 160-165)

Reviewer 1:

1. Line 36 of abstract missing apostrophe at participants’

RESPONSE: Amended.

2. Line 118 in Methods- you have gender when I think you mean sex (as your labels are male or female)

RESPONSE: Amended. 

3. Clear account of methodology although I’d welcome a bit more information on how the interview schedule was developed, and perhaps the schedule made available so future studies could ask the same questions.

RESPONSE: Apologies for this omission. We have added the following sentence to the revised manuscript and included the interview schedule in supplementary materials.

“The interview schedule was developed by authors DK and HD following agreement of the study research questions (see supplementary materials S1 for a copy of the interview schedule.” (lines 133-135).

4. Range of length of interview from 19 minutes to 47 seems very wide – any patterns in e.g. staff roles of the shorter interviews? Or any other reflection on this wide difference?

RESPONSE: When we checked the interview durations, we have realised there was a typo in the original manuscript. The longest interview was actually 56 minutes not 47 minutes although mean interview time does not change. This error in the manuscript has been corrected. Interview durations varied significantly, with shorter interviews (19–30 minutes) occurring across a range of roles, including supportive positions like Learning Support and Learning Mentor, as well as some senior roles like Deputy Head Teacher and Assistant Head Teacher. Potentially suggesting that the length of the interview was not related to the seniority of the position or work experience. However, longer interviews (46–56 minutes) were primarily associated with senior leadership positions such as Head Teachers and specialised roles like Outreach Support Worker. The shortest interview of 19 minutes was with a Deputy Head Teacher and this interview was cut short due to an issue in the PRU itself. To avoid the possibility of participants being able to identify themselves in the manuscript, we have not included these reflections in the revised manuscript. 

5. A bit more detail on the nature of the PRUs these staff work in would be helpful – what ages do they cater for? How many pupil places? Are they stand alone units? Some seem to be explicitly ‘therapeutic’ – what was the approach in others?

RESPONSE: We have included more information in the method section on the PRUs themselves in the revised manuscript. Please note that we were unable to find information on how pupil places were available in each PRU.

“Participants were from five PRUs and three different organisations. One PRU was a therapeutic school for young people of secondary school age supporting their mental health needs. The other four PRUs worked with children and young people excluded from primary and secondary mainstream schools and all offered short- and long-term placements.” (Lines 122-126)

6. Mentions the researcher engaging in conversations with the research team – some clarity on who did what would be useful (e.g. did one author conduct the interviews, transcribe them and analyse them?)

RESPONSE: We apologies for this lack of the detail. We have included the following sentences into the method section.

“To ensure that the quality and credibility of the results were upheld, all transcripts were analysed by the first author (DK) and independently checked by one of the co-authors (JH). First author (DK) was an MSc student at the time the study took place but was an experienced qualitative researcher and had previously worked in youth services. Author DK undertook the data analysis under the supervision of authors JK and HD.” (lines 160-165) 

7. L556 there’s a comment about may not be generalisable to other PRUs in different areas….that’s true – but the criteria considered given this type of study ought to be transferability (which will also be limited for the reasons given, but the use of ‘generalisability’ suggests a different kind of research (small-scale, in -depth, qualitative) – though I note the ‘essentialist’ approach to RFA which does point to a realist ontology so I know it isn’t a straightforward distinction, but I still found the ‘generalisability’ a bit jarring.

RESPONSE: We have amended this sentence to reflect the reviewer’s concerns.

“A key limitation of this study is that the interviews were conducted in a number of deprived areas across London and Southeast England. However, findings from this study should be broadly applicable to other PRU setting in comparable circumstances in other areas of the UK.” (lines 568-572)

Reviewer 2:

There are some unclear cases, for example:

The table is separated from its title:

RESPONSE: Amended

The sentence is missed in 251 line, isn't it? 

RESPONSE: Apologies but I am unclear as to what you mean. There does not appear to be a sentence missing. 

Whose participant (3 or 9) thoughts are there (426 line) share in the text?

RESPONSE: This quote is from participant 3 and we have removed the number 9 from the main text.

---

## [Editor Report · Decision Letter 1]

5 Sep 2024

Educator perceptions of the complex needs of young people in Pupil Referral Units: An Exploratory Qualitative Analysis

PONE-D-24-17686R1

Dear Dr. Dickson,

We’re pleased to inform you that your manuscript has been judged scientifically suitable for publication and will be formally accepted for publication once it meets all outstanding technical requirements.

Kind regards,

Cathryn Knight

Academic Editor

PLOS ONE

Additional Editor Comments (optional):

Thank you for addressing the feedback promptly and clearly. 
---

## [Editor Report · Acceptance letter]

11 Sep 2024

PONE-D-24-17686R1 

PLOS ONE

Dear Dr. Dickson, 

I'm pleased to inform you that your manuscript has been deemed suitable for publication in PLOS ONE. Congratulations! Your manuscript is now being handed over to our production team.

Kind regards, 

on behalf of

Dr. Cathryn Knight 

Academic Editor

PLOS ONE